# UNIFYING ALL SPECIES: LLM-BASED HYPER-HEURISTICS FOR MULTI-OBJECTIVE OPTIMIZATION

## ABSTRACT

Optimization problems are fundamental across various fields, including logistics, machine learning, and bioinformatics, where challenges are often characterized by complexity, high dimensionality. Modeling the interplay among multiple objectives is beneficial for optimization. However, existing Neural Combinatorial Optimization (NCO) methods and Large Language Model (LLM)-based approaches show limitations in adaptability and computational efficiency, primarily focusing on single-objective optimization. In this paper, we propose a novel framework, Multi-Objective Hierarchical Reflective Evolution (MHRE), for optimizing and generating heuristics algorithms for a broad range of optimization problems. Specifically, we extend the optimization space of the conventional hyper-heuristic methodologies, which allows us to unify similarity algorithms. We successfully construct Generalized Evolutionary Metaheuristic Algorithm (GEMA) for unifying metaheuristic algorithms. Yielding improved performance in experimental results. To show the performance of our method, we further applied the MHRE framework to optimize the Ant Colony Optimization (ACO) algorithm, achieving state-of-the-art results on random TSP problems and the TSPLib benchmark datasets. Our findings illustrate that the MLHH framework offers a robust and innovative solution for tackling complex optimization challenges, paving the way for future research in this area. For better reproducibility, we open source the code at `https://anonymous.4open.science/r/MHRE-BB53`.

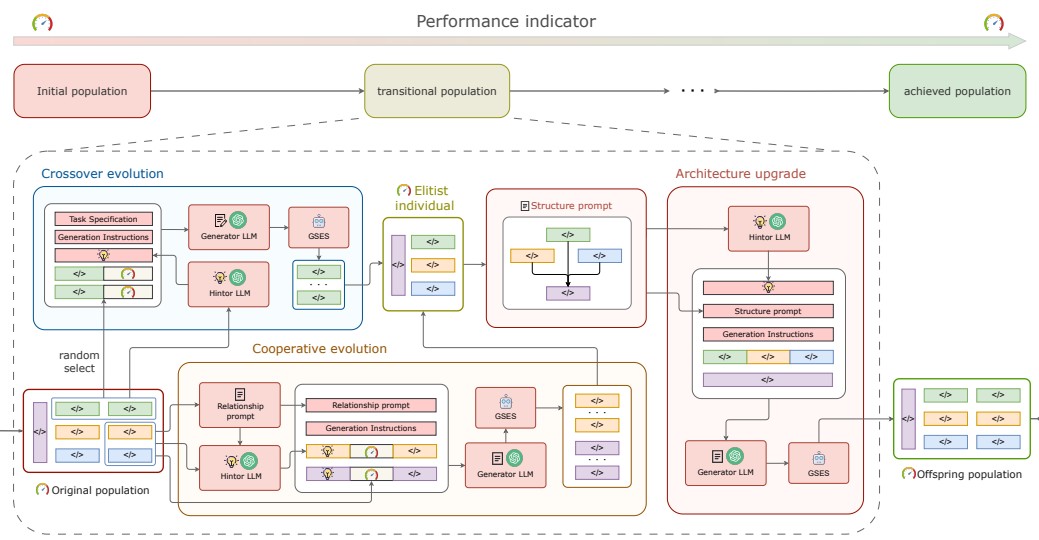

Figure 1: Overview of the Multi-Objective Hierarchical Reflective Evolution (MHRE) framework. Illustrating its hierarchical structure and how large language models (LLMs) are employed to optimize both sub-functions and architecture functions. The framework integrates Crossover Evolution, Cooperative Evolution, and Architecture Upgrade. To further standardize and improve optimization efficiency, we propose the Generation-Standardization-Evaluation-Selection (GSES) cycle.

## 1 INTRODUCTION

Optimization problems are fundamental across various fields, including logistics, machine learning, and bioinformatics, where challenges are often characterized by complexity, high dimensionality, and conflicting objectives. The rise of Neural Combinatorial Optimization (NCO) has introduced deep learning techniques into the optimization landscape, allowing models like Pointer Networks (Vinyals et al., 2015) and Graph Neural Networks (GNNs) (Khalil et al., 2017) to learn heuristics directly from data. These NCO methods have successfully solved classical NP-hard problems such as the Traveling Salesman Problem (TSP) and Vehicle Routing Problem (VRP) (Joshi et al., 2021), automating the process of heuristic discovery. However, NCO models are computationally expensive and require domain-specific customization, making them less adaptable to diverse problem types without significant retraining.

Recently, the emergence of Large Language Models (LLMs), such as GPT-4 (OpenAI, 2023), has opened new possibilities for optimization. LLMs, with their vast knowledge of heuristic strategies, can generate and refine optimization techniques, positioning them as powerful hyper-heuristic optimizers. Unlike traditional heuristics that are typically tailored for specific problems, hyper-heuristics generated by LLMs aim to create or select heuristics applicable across a broad spectrum of tasks (Burke et al., 2013). This shift from specialized solutions to flexible, general-purpose frameworks has gained significant attention due to its potential to streamline optimization processes across various domains. The flexibility of LLMs comes from their pre-training on vast datasets, enabling them to generate novel strategies without domain-specific fine-tuning. This flexibility minimizes manual tuning, making LLMs more efficient for solving complex, high-dimensional problems by dynamically adjusting strategies in real-time.

The existing approaches have leveraged LLMs as solvers or tools to enhance traditional metaheuristics. In the OPRO framework (Yang et al., 2024), LLMs are employed as black-box solvers, while in ReEvo (Ye et al., 2024), LLMs fine-tune heuristic of metaheuristic to improve performance. These methods, although promising, primarily focus on single-objective optimization, leaving the potential of LLMs for solving more complex, multi-objective optimization problems underexplored. Despite some advancements in expanding the optimization space for metaheuristics, the scope and versatility of these methods remain limited, particularly when addressing diverse and complex optimization challenges.

To bridge this gap, we aim to explore the extension of Language Hyper-Heuristics (LHHs) to multi-objective optimization problems (MOPs). However, directly applying existing LHH frameworks to multi-objective problems may be unsuitable for several reasons: **Firstly**, multiple heuristic functions may interact competitively, cooperatively, or hierarchically, and neglecting these dynamics could lead to suboptimal outcomes. **Secondly**, changes in one heuristic function can impact the adaptability of others, potentially degrading overall system performance.

To cope with the above problems, we introduce Multi-Objective Hierarchical Reflective Evolution (MHRE), a novel framework that extends Language Hyper-Heuristics (LHHs) to multi-objective optimization problems (MOPs) for enhancing the efficacy of heuristic solving while allowing for the unification of algorithmic architectures. Thereby broadening the optimization space and augmenting the effectiveness of heuristic solving. Unlike existing frameworks, MHRE is designed to achieve a unification of heuristic algorithms, optimizing and generating heuristics that are adaptable across various types of optimization tasks. MHRE operates by evolving a population of heuristics in a hierarchical process, leveraging LLMs not only to optimize individual sub-functions but also to dynamically adjust the overarching architecture of the optimization process.

Our work brings the following contributions:

- **Multi-Objective Language Hyper-Heuristics (MLHH):** We propose the concept of Multi-Objective Language Hyper-Heuristics (MLHH), To the best of our knowledge, this is the first framework that explores the potential of LHHs in solving multi-objective optimization problems.

- **The MHRE Framework:** We propose the Multi-Objective Hierarchical Reflective Evolution (MHRE) framework and demonstrate its efficacy within the Generalized Metaheuristic Framework (GEMA), achieving the goal of "unifying all species" in optimization.

- **Application to MHRE-ACO:** We enhance the Ant Colony Optimization (ACO) algorithm using the MHRE framework, achieving state-of-the-art results on random TSP problems and TSPLib benchmark datasets.

- **Experimental Validation:** Our experiments show that MHRE significantly improves optimization efficiency across diverse multi-objective problems. Specifically, the integration of Crossover Evolution, Cooperative Evolution, and Architecture Upgrade yields superior performance metrics compared to traditional approaches, demonstrating the robustness and scalability of the MHRE framework in addressing complex optimization challenges.

## 2 RELATED WORK

### 2.1 METAHEURISTIC ALGORITHMS AND COMBINATORIAL OPTIMIZATION

Metaheuristic algorithms have become essential for solving complex combinatorial optimization problems due to their efficiency in exploring vast search spaces. Classic examples include Genetic Algorithms (GA), Simulated Annealing (SA), Particle Swarm Optimization (PSO), and Ant Colony Optimization (ACO). ACO, in particular, has demonstrated substantial success in solving NP-hard problems like the Traveling Salesman Problem (TSP) and the Capacitated Vehicle Routing Problem (CVRP) by mimicking the foraging behavior of ants in nature. Solutions are built incrementally, guided by pheromone trails that represent learned information about the search space, which is updated iteratively based on solution quality (Dorigo & Gambardella, 1996; Kennedy & Eberhart, 1995; Kirkpatrick & Vecchi, 1983). Despite their success, these methods often require substantial manual tuning, limiting their generalization across diverse tasks (Talbi, 2009). To address these challenges, more adaptable methods that automate the selection and modification of heuristics have been developed, yet issues in flexibility and generalization persist (Boussaid et al., 2013; Yang, 2010).

### 2.2 LARGE LANGUAGE MODELS (LLMS) FOR HEURISTIC ALGORITHM SOLVING

LLMs have emerged as powerful tools not only for text-based tasks but also for solving complex optimization problems. The OPRO framework, for example, demonstrated the potential of LLMs as solvers or tools to enhance traditional metaheuristics (Yang et al., 2024). The ReEvo framework applies LLMs to enhance the performance of traditional metaheuristics such as ACO, PSO, and GA, unifying them under a single adaptive framework (Ye et al., 2024). By learning from vast datasets, LLMs can model the structure of optimization problems and predict near-optimal solutions, bypassing the need for handcrafted heuristics (Bengio et al., 2020). LLMs have also been used as hyper-heuristics to dynamically adjust the parameters of traditional algorithms, improving their performance across multiple problem domains (Ye et al., 2024; Durasevic & Jakobovic, 2020). This approach significantly reduces the need for manual intervention while improving performance on tasks like TSP and CVRP (Khalil et al., 2017).

### 2.3 HYPER-HEURISTICS IN COMBINATORIAL OPTIMIZATION

Traditional hyper-heuristics provide a generalized framework for automating the selection or generation of low-level heuristics, reducing the reliance on expert-designed components (Burke et al., 2013). However, these methods still often require manually crafted components, which limits their ability to generalize to novel problems (Sabar et al., 2013). LLMs present a promising solution to this limitation by learning generalized representations from large-scale data, enabling the automated design of heuristic rules. Integrating LLMs into hyper-heuristic design allows for greater flexibility and improved performance without manual tuning (Ye et al., 2024; Yang et al., 2024). This integration is poised to play a key role in the future of combinatorial optimization, automating algorithm design while maintaining high efficiency and adaptability.

## 3 LANGUAGE HYPER-HEURISTICS FOR MULTI-OBJECTIVE OPTIMIZATION

Hyper-heuristics (HHs) are high-level search methodologies that explore a space of heuristics to select or generate effective strategies for solving underlying optimization problems. In multi-objective

optimization problems (MOPs), HHs aim to find heuristics that effectively approximate the Pareto front, thereby optimizing multiple conflicting objectives simultaneously. This dual-level framework is formally defined as follows:

**Definition 1** (Hyper-Heuristic for MOP). *Given a multi-objective optimization problem with solution space $S$ and objective vector function $\mathbf{f} : S \to \mathbb{R}^k$, a hyper-heuristic searches for an optimal heuristic $h^*$ in a heuristic space $H$ that minimizes a meta-objective function $F : H \to \mathbb{R}$:*

$$h^* = \arg \min_{h \in H} F(h),$$

*where*

$$F(h) = \Phi\left(\mathbf{f}(S_h)\right),$$

*and $S_h \subseteq S$ is the set of solutions generated by heuristic $h$, and $\Phi$ is a performance indicator measuring the quality of $\mathbf{f}(S_h)$ in approximating the Pareto front.*

Traditional HHs are often categorized into heuristic selection and heuristic generation approaches, relying on manually defined heuristic components or rules. However, these methods may be limited by the predefined heuristic space $H$ and might not capture the full potential of novel heuristics.

To address these limitations, we introduce *Language Hyper-Heuristics with Multi-Objective Hierarchical Reflective Evolution* (MHRE), a framework that leverages Large Language Models (LLMs) to generate heuristics within an open-ended heuristic space. MHRE enhances the exploration of complex heuristic spaces by employing a hierarchical and cooperative evolutionary process involving sub-functions and architecture functions.

### 3.1 Multi-Objective Hierarchical Reflective Evolution

The MHRE framework operates by evolving a population of heuristics composed of two distinct types of functions:

- **Sub-functions** ($\mathcal{F}$): Specialized functions responsible for specific tasks within the heuristic algorithm, such as performing local searches or implementing problem-specific operations.

- **Architecture Functions** ($\mathcal{A}$): High-level functions that integrate information from sub-functions to make core decisions within the heuristic, handling parameter tuning and adapting the overall strategy based on feedback from the optimization process.

By structuring heuristics into sub-functions and architecture functions, MHRE enables a hierarchical and cooperative evolutionary process that explores complex heuristic spaces more effectively than traditional HHs or LHHs alone.

### 3.2 Formal Definition of MHRE for MOPs

**Definition 2** (MHRE for MOP). *Given a multi-objective optimization problem with solution space $S$ and objective vector function $\mathbf{f} : S \to \mathbb{R}^k$, the MHRE framework searches for an optimal set of heuristics $H^* \subseteq H$, where $H$ consists of combinations of sub-functions $\phi \in \mathcal{F}$ and architecture functions $\alpha \in \mathcal{A}$. The goal is to minimize the meta-objective function $F : H \to \mathbb{R}$:*

$$H^* = \arg \min_{H \subseteq \mathcal{F} \times \mathcal{A}} F(H),$$

*where*

$$F(H) = \frac{1}{|H|} \sum_{h \in H} \Phi\left(\mathbf{f}(S_h)\right),$$

*and for each heuristic $h = (\phi, \alpha) \in H$, $S_h \subseteq S$ is the set of solutions generated by $h$, and $\Phi$ is a performance indicator that measures how well $\mathbf{f}(S_h)$ approximates the Pareto front.*

By leveraging LLMs to generate and refine both sub-functions and architecture functions, MHRE explores a vast and diverse heuristic space, potentially discovering novel and effective heuristics beyond human-designed components. The hierarchical cooperative evolution in MHRE allows for complex interactions between heuristic components, enhancing the capability to solve intricate MOPs effectively.

# 4 LANGUAGE HYPER-HEURISTICS WITH MULTI-OBJECTIVE HIERARCHICAL REFLECTIVE EVOLUTION

Building upon the concept of Language Hyper-Heuristics (LHHs), we introduce a novel framework tailored for multi-objective optimization problems (MOPs), termed *Multi-Objective Hierarchical Reflective Evolution* (MHRE). Unlike traditional approaches that may focus solely on a single optimization problem, MHRE is designed to optimize and generate heuristics applicable to a broad range of optimization problems. The framework incorporates two distinct types of functions—*sub-functions* and *architecture functions*—and leverages Large Language Models (LLMs) to facilitate a cooperative evolutionary process aimed at discovering effective heuristics.

## 4.1 OPTIMIZATION FLOW STEPS

MHRE operates by evolving a population of heuristics, each composed of sub-functions and architecture functions. The optimization flow of MHRE consists of three main steps, each designed to enhance different aspects of the heuristic population through cooperative evolution:

**Co-evolution of Same-Type Functions**    In this initial step, functions of the same type undergo crossover evolution. The hinter LLM analyzes a randomly selected pair of functions—one superior and one inferior—and provides optimization suggestions based on their differences. These suggestions are then used by the generator LLM to produce new individuals, enhancing the population with improved function variants.

**Co-evolution of Different-Type Functions**    Next, functions of different types engage in cooperative evolution. The hinter LLM receives a sub-function and an architecture function, and through relational analysis, it generates optimization suggestions that enhance their interaction. The generator LLM utilizes these insights to create new functions that better cooperate, leading to heuristics with improved performance.

**Framework Upgrade**    Finally, the framework undergoes an upgrade based on elite individuals in the population. The hinter LLM analyzes the top-performing architecture functions, offering optimization suggestions for refinement. The generator LLM then produces upgraded architecture functions.

## 4.2 GENERATION-STANDARDIZATION-EVALUATION-SELECTION (GSES) CYCLE

To systematically refine the heuristic population in each iteration, we employ the Generation-Standardization-Evaluation-Selection (GSES) cycle in Figure 2. This cycle encompasses four key steps—generation, standardization, evaluation, and selection—that work together to enhance the quality and performance of the heuristics within the framework.

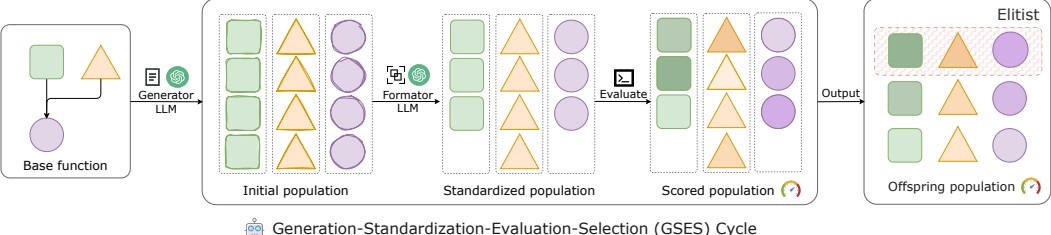

Generation-Standardization-Evaluation-Selection (GSES) Cycle

Figure 2: Overview of the Generation-Standardization-Evaluation-Selection (GSES) cycle. This cycle iteratively refines the heuristic population in the MHRE framework. The generator LLM creates a new population, which is then standardized by the Formator LLM to correct inconsistencies. The standardized functions are evaluated, and top-performing ones are selected based on their performance alongside elite individuals. This process ensures continuous improvement in heuristic quality and effectiveness.

Table 1: Behavioral Correspondence Among Metaheuristic Algorithms. This table shows the presence or absence of key behaviors (Local Search, Global Search, Following Behavior, and Mutation Behavior) across different metaheuristic algorithms. The identified patterns serve as the foundation for the unified optimization approach in the MHRE framework, allowing these behaviors to be abstracted and generalized for more adaptable and efficient heuristics.

| Algorithm | Local Search | Global Search | Following | Mutation |
|-----------|--------------|---------------|-----------|----------|
| AFSA | ✓ | ✗ | ✓ | ✗ |
| CSA | ✗ | ✓ | ✗ | ✓ |
| FLA | ✗ | ✗ | ✓ | ✓ |
| WOA | ✗ | ✓ | ✗ | ✗ |
| PSO | ✓ | ✓ | ✓ | ✗ |

*Note:* The checkmarks (✓) indicate that the algorithm exhibits the corresponding behavior, while crosses (✗) indicate absence of the behavior.

Each iteration of the evolutionary process follows this consistent procedure to generate offspring populations. Initially, the generator LLM produces an initial population of heuristic functions. However, some individuals may be unsuitable for direct use due to issues such as input/output format discrepancies or inconsistent parameter naming. To address this, we introduce an intermediate step utilizing a *Formator LLM*, which standardizes the generated functions by correcting minor flaws. Severe issues, such as parameter anomalies or data mismatches, lead to the deletion of the affected individuals, resulting in a standardized population ready for evaluation.

Subsequently, we evaluate the standardized population on a test set to obtain performance scores for each individual function. Importantly, the score of each function is derived based on its performance in conjunction with the elite individuals from other function groups, reflecting the cooperative nature of the heuristic components within the framework. Following evaluation, we rank the individuals and perform selection to curate a new population with a specified number of individuals. The top-performing functions are designated as elite individuals, ensuring that their advantageous traits are preserved for future iterations. This GSES cycle iterates to progressively enhance the overall quality of the heuristic population, systematically refining the heuristics through generation, standardization, evaluation, and selection.

## 5 EXPERIMENTS

In this section, we assess the effectiveness of the proposed Multi-Objective Hierarchical Reflective Evolution (MHRE) framework through two sets of experiments. The first experiment focuses on demonstrating MHRE's ability to unify and optimize multiple metaheuristic algorithms by identifying and leveraging common behavioral patterns. The second experiment applies the MHRE framework specifically to the Ant Colony Optimization (ACO) algorithm, showcasing its potential to enhance and refine existing optimization techniques. These experiments collectively validate the framework's capacity to improve both efficiency and adaptability across diverse optimization problems.

### 5.1 UNIFYING METAHEURISTIC ALGORITHMS

Metaheuristic algorithms, such as the Artificial Fish Swarm Algorithm (AFSA), Cuckoo Search Algorithm (CSA), Frog Leaping Algorithm (FLA), and Whale Optimization Algorithm (WOA), exhibit core similarities in their mechanisms despite differences in agent behavior. These algorithms focus on exploring solution spaces, optimizing candidates, and converging towards optimal solutions. By analyzing these algorithms, we identified four fundamental behavioral patterns: Local Search, Global Search, Following Behavior, and Mutation Behavior.

Through iterative reflection, we abstracted these principles into generalized functions that capture the essence of these behaviors. This abstraction allows us to create a unified optimization approach, streamlining the design of metaheuristics while improving performance.

Table 2: Performance Comparison of GEMA and Other Metaheuristic Algorithms on Benchmark Functions. This table compares the performance of GEMA (proposed framework) with traditional metaheuristic algorithms (AFSA, WOA, CSA, PSO, FLA) across several standard benchmark functions.

| Benchmark | GEMA(ours) | AFSA | WOA | CSA | PSO | FLA |
|---|---|---|---|---|---|---|
| Sphere | **0.008** | 1.280 | **0.000** | 122.479 | 4.082 | 8.299 |
| Rastrigin | 0.123 | 46.996 | **0.000** | 365.123 | 162.679 | 378.326 |
| Ackley | 2.591 | 2.564 | **0.000** | 8.208 | 2.552 | 3.782 |
| Griewank | **0.007** | 0.101 | **0.000** | 1.030 | 0.384 | 0.908 |
| Levy | **0.823** | 1.347 | 10.679 | 35.250 | 5.979 | 31.279 |
| Schwefel | **0.111** | 12451.214 | 12451.214 | 12483.42 | 1245.099 | 12551.835 |
| Rosenbrock | **0.272** | 269.036 | 28.737 | 88152.181 | 242.587 | 1362.773 |
| Michalewicz | -6.35 | -10.984 | -14.373 | -7.648 | -12.659 | **-4.092** |
| Zakharov | **0.634** | 6.075 | 239.083 | 233.855 | 21.825 | 48.269 |
| Alpine | 1.27 | 1.892 | **0.000** | 30.382 | 2.041 | 20.958 |

*Note: Results closer to 0 indicate better performance.

- **Local Search:** This pattern involves focused exploration around a given agent to fine-tune potential solutions within a local area. For instance, the foraging behavior in AFSA and the local search mechanisms in PSO (Particle Swarm Optimization) are driven by this principle.

- **Global Search:** In contrast to local search, global search emphasizes broad exploration across the entire solution space to avoid premature convergence to suboptimal solutions. This is exemplified by the Lévy flight mechanism in CSA and the global best guidance in PSO.

- **Following Behavior:** Here, agents adjust their positions by mimicking better-performing individuals in the population. Examples include the following mechanism in AFSA and the frog leaping towards better local optima in FLA.

- **Mutation Behavior:** This behavior introduces randomness to increase diversity and help escape local optima, thereby preventing premature convergence. Random jumps in FLA and the nest replacement in CSA illustrate this type of behavior.

The MHRE framework encapsulates these common behaviors into generalized sub-functions and architecture functions. The multi-objective optimization process iteratively refines these functions, producing robust and adaptable heuristics that are optimized for a wide range of problems. This approach offers a novel unification strategy, surpassing the limitations of traditional metaheuristic design by integrating and optimizing common patterns across multiple algorithms.

In this experiment, we conduct a comprehensive evaluation of the proposed Generalized Evolutionary Metaheuristic Algorithm (**GEMA**) across standard benchmark functions. GEMA consistently outperformed other metaheuristic algorithms, not only in convergence speed but also in adaptability to high-dimensional search spaces.

Traditional metaheuristics, such as AFSA, WOA, CSA, PSO, and FLA, are limited by their reliance on specific natural behaviors, restricting their generality across diverse problem domains. GEMA, on the other hand, introduces a unified evolutionary framework that integrates local and global search strategies, providing superior performance on both simple and complex benchmark functions.

The results demonstrate GEMA's strong ability to navigate non-linear, high-dimensional landscapes, outperforming traditional algorithms in robustness and convergence. These findings highlight GEMA's potential as a generalized optimization method applicable to a wide range of tasks, significantly extending the scope of metaheuristic applications.

In conclusion, GEMA provides a flexible and robust solution to diverse optimization tasks, addressing the long-standing issue of homogeneity in metaheuristic algorithms and paving the way for more adaptable optimization methods in future research.

## 5.2 Main Exepriments

In this section, we evaluate the effectiveness of the proposed *Multi-Objective Hierarchical Reflective Evolution* (MHRE) framework through a series of experiments aimed at optimizing Ant Colony

Optimization (ACO) for combinatorial optimization problems (COPs). Specifically, we compare the performance of the MHRE-ACO algorithm with existing state-of-the-art algorithms across multiple problem sizes, including instances from the well-known TSPLIB dataset G. Reinelt (1991). These experiments serve to assess both the scalability and adaptability of the MHRE framework in handling increasingly complex optimization tasks.

### 5.2.1 SETUP

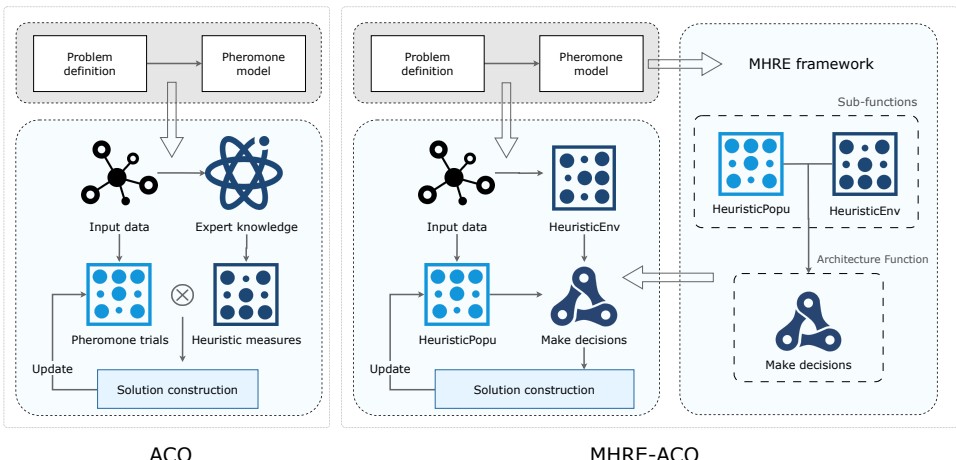

Figure 3: Comprehensive Overview of the MHRE-ACO Algorithm: Integrating Multi-Objective Hierarchical Reflective Evolution to Enhance Ant Colony Optimization for Combinatorial Problems.

To assess the effectiveness of the *Multi-Objective Hierarchical Reflective Evolution* (MHRE) framework, we conducted experiments focusing on optimizing Ant Colony Optimization (ACO) heuristics for a variety of combinatorial optimization problems (COPs). Figure **??** illustrates the MHRE-ACO framework. In this framework, the ACO algorithm is iteratively refined using MHRE's reflective evolutionary process. This process enables the dynamic adjustment of search strategies, including local and global search behaviors, thereby enhancing the overall optimization capabilities of ACO.

The MHRE-ACO algorithm was evaluated in two distinct experimental settings. First, it was tested on randomly generated TSP instances with varying problem sizes (e.g., TSP with 100, 500, and 1000 cities) to assess its scalability and generalization capabilities across different problem complexities. In the second phase, the algorithm was applied to a subset of TSPLIB (G. Reinelt, 1991), including well-established benchmark instances, to evaluate its optimization performance and compare it against state-of-the-art algorithms in structured, real-world problem environments.

Key parameters such as population size, number of generations, and the number of function evaluations were kept consistent across all experiments to ensure fairness. Each experiment was repeated 10 times to account for statistical variability, with the average performance across runs being used for comparison.

### 5.2.2 RESULTS

The experimental results, first demonstrated in Table 3, highlight the strong performance of the *Multi-Objective Hierarchical Reflective Evolution* (MHRE) framework on randomly generated TSP instances, particularly in larger problem sizes like TSP1000. MHRE-ACO consistently outperformed competing algorithms, achieving solutions that closely approached those of the SOTA solver. This success is largely due to MHRE's ability to dynamically balance exploration and exploitation through its reflective evolutionary mechanisms, allowing it to efficiently navigate complex, high-dimensional solution spaces.

Furthermore, as shown in Table 4, MHRE's performance on the structured TSPLIB dataset further validates its robustness and scalability. The framework consistently achieved higher solution quality compared to other methods across all tested instances. In larger, more challenging TSPLIB

Table 3: Performance Evaluation of Algorithms on Randomly Generated TSP Instances (TSP20 to TSP1000), with Partial Data Referenced from AEL (Algorithm Evolution Using Large Language Models) (Smith et al., 2024).

| Problem Size | SOTA Solver LKH3 | Human (Greedy) | MHRE+ACO (ours) | ReEvo +ACO | Constructive | AEL (GPT-4) |
|---|---|---|---|---|---|---|
| 20 | 3.84 | 4.49 | **3.64** | 3.85 | 5.34 | 4.07 |
| 50 | 5.69 | 7.01 | **5.63** | 5.76 | 8.19 | 6.33 |
| 100 | 7.77 | 9.84 | **8.06** | 8.18 | 11.3 | 8.58 |
| 500 | 16.56 | 20.87 | **18.09** | 20.05 | 22.76 | 18.67 |
| 1000 | 23.08 | 28.9 | 26.21 | 30.4 | 31.1 | **26.03** |

Table 4: Results on Subsets of TSPLib. The last column represents the optimal solution that has been found in this task. Each cell shows a function score representing the result of the algorithm optimization with a ratio to the optimal score in parentheses. Cells without value indicate unsuccessful attempts at completing the task.

| Task | ReEvo+ACO | DeepACO (n=100) | DeepACO (n=500) | MHRE+ACO (ours) | Optimal |
|---|---|---|---|---|---|
| a280 | 2942 (14.07%) | 3160 (22.55%) | 3156 (22.39%) | **2924 (13.39%)** | 2579 |
| att48 | 34984 (4.36%) | 34369 (2.53%) | 34938 (4.22%) | **34046 (1.56%)** | 33522 |
| att532 | 97427 (12.34%) | 118691 (36.85%) | 117044 (34.95%) | **97329 (12.22%)** | 86729 |
| ch130 | 6528 (6.85%) | 6727 (10.09%) | 6535 (6.96%) | **6377 (4.38%)** | 6110 |
| ch150 | 6794 (4.07%) | 7078 (8.43%) | 7276 (11.45%) | **6779 (3.85%)** | 6528 |
| d1291 | 58678 (15.5%) | 138128 (171.9%) | 102817 (102.39%) | **55113 (8.49%)** | 50801 |
| d1655 | 74098 (19.27%) | - | - | **68619 (10.45%)** | 62128 |
| d198 | 17463 (10.66%) | 20986 (32.99%) | 19166 (21.46%) | **15822 (0.27%)** | 15780 |
| d493 | 39044 (11.55%) | 50834 (45.23%) | 46619 (33.19%) | **35019 (0.05%)** | 35002 |
| d657 | 56346 (15.2%) | 76611 (56.63%) | 73884 (51.06%) | **54101 (10.61%)** | 48912 |
| eil101 | 678 (7.72%) | 673 (7.02%) | **670 (6.49%)** | 675 (7.29%) | 629 |
| eil51 | 436 (2.27%) | 543 (27.37%) | 437 (2.49%) | **432 (1.36%)** | 426 |
| eil76 | 561 (4.31%) | 562 (4.45%) | 567 (5.33%) | **556 (3.41%)** | 538 |
| fl1400 | 24719 (22.81%) | - | 99209 (392.92%) | **23684 (17.67%)** | 20127 |
| fl1577 | 25785 (15.89%) | - | 71870 (223.03%) | **24795 (11.44%)** | 22249 |
| fl417 | **13671 (15.26%)** | 51267 (332.23%) | 25164 (112.16%) | 13794 (16.3%) | 11861 |
| gil262 | **2608 (9.66%)** | 2663 (11.97%) | 2727 (14.66%) | 2613 (9.88%) | 2378 |
| kroA100 | 22709 (6.7%) | 24433 (14.81%) | 24792 (16.49%) | **22575 (6.07%)** | 21282 |
| kroA150 | 29158 (9.93%) | 30916 (16.56%) | 31458 (18.6%) | **28917 (9.02%)** | 26524 |
| kroA200 | 32482 (10.6%) | 35260 (20.06%) | 35208 (19.89%) | **31590 (7.56%)** | 29368 |
| kroB100 | 23571 (6.46%) | 24412 (10.26%) | 24846 (12.22%) | **22779 (2.88%)** | 22141 |
| kroB150 | 29209 (11.78%) | 30327 (16.06%) | 30482 (16.65%) | **29048 (11.17%)** | 26130 |
| kroB200 | 33181 (12.72%) | 35291 (19.89%) | 34733 (17.99%) | **32049 (8.87%)** | 29437 |
| kroC100 | 22082 (6.42%) | 23684 (14.14%) | 24784 (19.44%) | **21800 (5.06%)** | 20749 |
| kroD100 | 22615 (6.2%) | 23803 (11.78%) | 23917 (12.32%) | **22481 (5.57%)** | 21294 |
| vm1084 | 284951 (19.08%) | 905479 (278.39%) | 532173 (122.39%) | **281503 (17.64%)** | 239297 |

problems, MHRE's adaptability and refined search processes were key factors in its superior performance, allowing it to closely approximate optimal solutions while maintaining computational efficiency.

The scalability of MHRE was another notable advantage. As the problem size increased, MHRE maintained its efficiency, consistently converging to high-quality solutions. In contrast, ReEvo, while effective in smaller instances, showed a noticeable decline in both efficiency and solution quality as the complexity of the problem grew. MHRE's hierarchical and reflective evolutionary processes allowed it to handle the increased complexity with minimal performance degradation.

As shown in Table 3, MHRE+ACO consistently delivered superior performance compared to other approaches, including ReEvo+ACO and human-designed greedy algorithms. Particularly in larger problem sizes like TSP1000, MHRE+ACO maintained efficiency and scalability, offering improved convergence over traditional methods.

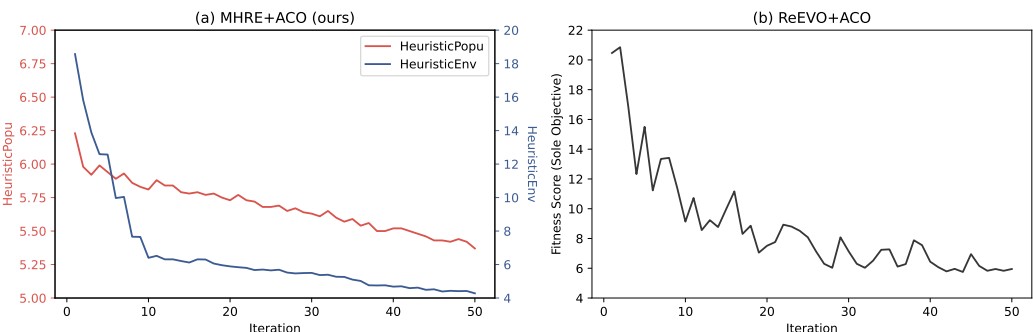

Figure 4: Convergence Curve Comparison over Iterations. The MHRE+ACO algorithm consistently converges faster to near-optimal solutions across all problem sizes.

Figure 4 further illustrates the rapid convergence of MHRE+ACO compared to other algorithms. MHRE's adaptive mechanisms enabled it to efficiently navigate the solution space, reaching optimal solutions with fewer iterations. This improved convergence is particularly evident in larger instances such as TSP1000, where MHRE+ACO consistently demonstrated faster and more stable results.

## 5.3 ABLATION STUDY

We conduct extra experiments on the utility of different components in MHRE. The experiments show that *Crossover Evolution* provides a foundational optimization mechanism, the integration of *Cooperative Evolution* and *Architecture Upgrade* substantially boosts the model's performance. Details are presented in Appendix A.

## 6 CONCLUSION

In this work, we introduced the Multi-Objective Language Hyper-Heuristics (MLHH) framework, which significantly advances the field of multi-objective optimization. Our contributions include the proposal of the MHRE framework, which successfully integrates and optimizes multiple meta-heuristic algorithms, demonstrating the effectiveness of unifying different optimization strategies.

Through comprehensive ablation experiments, we validated the individual and combined impacts of the three key components: Crossover Evolution, Cooperative Evolution, and Architecture Upgrade. The results indicated that while Crossover Evolution provides a solid foundation for optimization, the addition of Cooperative Evolution markedly enhances the efficiency of weaker functions, especially when dealing with inconsistent performance. Furthermore, the Architecture Upgrade component allows for further improvements in the model's upper-performance limits.

Overall, the MLHH framework not only offers an innovative approach to tackling multi-objective optimization problems but also sets the stage for future research to explore the potential of combining various heuristic strategies for improved algorithmic performance.

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

## A   DETAILS OF ABLATION STUDY ON COMPONENTS' UTILITY

To evaluate the contributions of the three main components of our model (i.e., Crossover Evolution, Cooperative Evolution, and Architecture Upgrade), we conducted a series of experiments on TSP. We recorded data from three distinct experimental setups: using only Crossover Evolution, combining Crossover Evolution with Cooperative Evolution, and employing all three components together. The results are shown in Table 5.

Table 5: Performance Evaluation of MHRE with Different Components

| Method | TSP20 | TSP50 | TSP100 | TSP500 | TSP1000 |
|---|---|---|---|---|---|
| Only Crossover | 3.76 | 5.88 | 8.41 | 19.12 | 28.45 |
| w/ Cooperative | 3.76 | 5.75 | 8.19 | 18.61 | 26.88 |
| Full MHRE | **3.64** | **5.63** | **8.06** | **18.09** | **26.71** |

The first experiment focused solely on Crossover Evolution, which facilitates the crossover of heuristics among similar functions. The second experiment incorporated Cooperative Evolution, allowing for the combination of dissimilar functions. The results indicated that when the performance of the

functions was inconsistent, Cooperative Evolution significantly enhanced the optimization efficiency of the weaker function objectives compared to Crossover Evolution alone.

Furthermore, we introduced the Architecture Upgrade component in the third experiment, which yielded an additional improvement in the upper performance limits of the model. The incorporation of this component demonstrates the synergistic effect of combining all three elements, leading to superior overall results.

In summary, the experiments illustrate that while Crossover Evolution provides a foundational optimization mechanism, the integration of Cooperative Evolution and Architecture Upgrade substantially boosts the model's performance, particularly in scenarios where function performance varies significantly.

## B  GEMA EXPERIMENT: ARCHITECTURE AND SUB-FUNCTIONS

This section presents the architecture function and four sub-functions used in GEMA. We provide the initial prompt for each function and the corresponding seed function that was used.

### B.1  ARCHITECTURE FUNCTION: UPDATE_POSITION

```
Design an architecture function named update_position that balances local search, global search, mutation,
    and following behavior to optimize the individual's position.
```

Listing 1: System prompt for architecture function (update_position).

```python
import numpy as np
from mutate import mutate
from follow import follow
from global_search import global_search
from local_search import local_search

def update_position(self, individual):
    local_step = 0.5
    local_prob = 0.3
    global_step = 0.5
    global_prob = 0.4
    follow_prob = 0.2
    mutation_prob = 0.1
    mutation_step = 0.6
    alpha = 0.6
    beta = 0.5

    r = np.random.rand()

    if r < local_prob:
        local_search(individual, local_step, self.lower_bound, self.upper_bound, self.dim)

    elif r < local_prob + global_prob:
        global_search(individual, global_step, alpha, beta, self.best_individual, self.lower_bound,
            self.upper_bound)

    elif r < local_prob + global_prob + follow_prob:
        follow(individual, beta, self.population, self.lower_bound, self.upper_bound)

    elif r < local_prob + global_prob + follow_prob + mutation_prob:
        mutate(individual, mutation_step, self.lower_bound, self.upper_bound, self.dim)
```

Listing 2: Seed function for update_position.

### B.2  SUB-FUNCTION: FOLLOW

```
Design a follow function that adjusts the position of an individual by following another, more successful
    individual in the population.
```

Listing 3: System prompt for follow function.

```python
import numpy as np

def follow(individual: dict, beta: float, population: list, lower_bound: float, upper_bound: float) -> None:
    chosen_individual = np.random.choice(population)
    direction = chosen_individual['position'] - individual['position']
    norm = np.linalg.norm(direction)
    if norm > 1e-8:
```

```
step = beta * direction / norm
new_position = individual['position'] + step
individual['position'] = np.clip(new_position, lower_bound, upper_bound)
```

Listing 4: Seed function for follow.

### B.3 SUB-FUNCTION: MUTATE

```
Design a mutate function that introduces random variations in an individual's position to promote exploration
    and prevent premature convergence.
```

Listing 5: System prompt for mutate function.

```
import numpy as np

def mutate(individual: dict, mutation_step: float, lower_bound: float, upper_bound: float, dim: int) -> None:
    mutation = mutation_step * np.random.uniform(-1, 1, dim)
    individual['position'] = np.clip(individual['position'] + mutation, lower_bound, upper_bound)
```

Listing 6: Seed function for mutate.

### B.4 SUB-FUNCTION: GLOBAL_SEARCH

```
Design a global search function that moves an individual towards the best-known solution in the population.
```

Listing 7: System prompt for global_search function.

```
import numpy as np

def global_search(individual: dict, global_step: float, alpha: float, beta: float, best_individual_position:
    np.array, lower_bound: float, upper_bound: float) -> None:
    global_best_position = best_individual_position
    if global_best_position is not None:
        direction = global_best_position - individual['position']
        norm = np.linalg.norm(direction)
        if norm > 1e-8:
            step = global_step * alpha * direction / norm
            new_position = individual['position'] + step
            individual['position'] = np.clip(new_position, lower_bound, upper_bound)
```

Listing 8: Seed function for global_search.

### B.5 SUB-FUNCTION: LOCAL_SEARCH

```
Design a local search function that fine-tunes an individual's position by exploring its neighborhood to
    improve solution quality.
```

Listing 9: System prompt for local_search function.

```
import numpy as np

def local_search(individual: dict, local_step: float, lower_bound: float, upper_bound: float, dim: int) ->
    None:
    step = local_step * np.random.uniform(-1, 1, dim)
    new_position = individual['position'] + step
    individual['position'] = np.clip(new_position, lower_bound, upper_bound)
```

Listing 10: Seed function for local_search.

## C  MHRE-ACO EXPERIMENT: ARCHITECTURE AND SUB-FUNCTIONS

This section presents the architecture function and two sub-functions used in the MHRE-ACO experiment, along with their prompts and seed functions. These functions collaboratively contribute to optimizing the ant colony optimization (ACO) process.

## C.1 ARCHITECTURE FUNCTION: PICK_MOVE

```
Design a pick_move function that takes the heuristic outputs from the HeuristicPopu and HeuristicEnv
    functions and bases its action decision on the heuristic information provided by both.
```

Listing 11: System prompt for architecture function (pick_move).

```python
import torch
from torch.distributions import Categorical
from typing import Tuple, Optional

def pick_move(global_popu_weight: torch.Tensor, global_env_weight: torch.Tensor, prev: torch.Tensor, mask:
        torch.Tensor, require_prob: bool) -> Tuple[torch.Tensor, Optional[torch.Tensor]]:
    alpha = 1.0
    beta = 3

    popu_weight = global_popu_weight[prev] # shape: (n_agents, p_size)
    env_weight = global_env_weight[prev] # shape: (n_agents, p_size)

    popu_weight_log = torch.log1p(popu_weight)
    env_weight_log = torch.log1p(env_weight)

    weighted_sum = alpha * popu_weight_log + beta * env_weight_log
    weighted_sum *= mask

    probs = torch.softmax(weighted_sum, dim=1)

    dist = Categorical(probs=probs)
    actions = dist.sample() # shape: (n_agents,)
    log_probs = dist.log_prob(actions) if require_prob else None # shape: (n_agents,)

    return actions, log_probs
```

Listing 12: Seed function for pick_move.

## C.2 SUB-FUNCTION: HEURISTICENV

The 'HeuristicEnv' function computes heuristic estimates that reflect the potential benefit of each edge being part of the optimal tour in the optimization process.

```
Design a HeuristicEnv function that computes heuristic estimates for each edge, helping to determine which
    edges should be part of the optimal solution in the optimization problem.
```

Listing 13: System prompt for HeuristicEnv function.

```python
import torch

def HeuristicEnv(edge_attr: torch.Tensor) -> torch.Tensor:
    num_edges = edge_attr.size(0)
    heuristic_values = torch.zeros_like(edge_attr)

    transformed_attr = torch.log1p(torch.abs(edge_attr))

    mean = transformed_attr.mean(dim=0, keepdim=True)
    std = transformed_attr.std(dim=0, keepdim=True)
    edge_attr_norm = (transformed_attr - mean) / (std + 1e-7)

    heuristic_values = torch.exp(-8 * edge_attr_norm)
    heuristic_values[torch.isnan(heuristic_values)] = 0
    heuristic_values = torch.clamp(heuristic_values, min=0)

    return heuristic_values
```

Listing 14: Seed function for HeuristicEnv.

## C.3 SUB-FUNCTION: HEURISTICPOPU

```
Design a HeuristicPopu function that updates the global heuristic matrix based on the paths taken by agents
    and their associated costs, reflecting the significance of each edge.
```

Listing 15: System prompt for HeuristicPopu function.

```python
import torch

def HeuristicPopu(global_popu_weight: torch.Tensor, paths: torch.Tensor, costs: torch.Tensor) -> torch.Tensor:
    decay = 0.9
    n_agent = paths.size(0)
```

```
new_popu_weight = global_popu_weight * decay

path_usage = torch.zeros_like(new_popu_weight)

for i in range(n_agent):
    path = paths[i]
    cost = costs[i]
    path_usage[path, torch.roll(path, shifts=1)] += 1.0 / (cost + 1e-7)

path_fitness = 1.0 / (costs + 1e-7)
fitness_threshold = 1.0 / (torch.mean(costs) + 1e-7)

for j in range(n_agent):
    path = paths[j]
    fitness_score = path_fitness[j]

    if fitness_score > fitness_threshold:
        path_contribution = path_usage[path, torch.roll(path, shifts=1)].sum() * fitness_score

        path_penalty = (path_usage[path, torch.roll(path, shifts=1)] < 1).float()
        new_popu_weight[path, torch.roll(path, shifts=1)] += path_contribution - path_penalty

new_popu_weight = new_popu_weight * 0.9 + global_popu_weight * 0.1
new_popu_weight = torch.clamp(new_popu_weight, min=0)

return new_popu_weight
```

Listing 16: Seed function for HeuristicPopu.

## C.4 RELATIONSHIP BETWEEN HEURISTICENV AND HEURISTICPOPU

```
The functions `HeuristicEnv` and `HeuristicPopu` work together to generate heuristic matrices. `HeuristicEnv`
    computes the environmental heuristic estimates for each edge, while `HeuristicPopu` updates the global
    heuristic matrix based on the population's path data and costs. Together, they balance the
    environmental and population information to optimize the overall routing strategy.
```

Listing 17: Relationship Between HeuristicEnv and HeuristicPopu

# D COMMON PROMPTS FOR LLMS

This section presents the common system and user prompts used for various Large Language Model (LLM) interactions, including checking function validity, providing optimization hints, and generating heuristic functions.

## D.1 SYSTEM PROMPT: CHECK_LLM

```
You are responsible for evaluating a Python function. Your task is to verify if the function strictly adheres
    to the provided input/output formats and matches the given sample data.

- If the function does not conform to the input format or fails to correctly run, return the string 'error'
    followed by a brief explanation of the issue.
- If the function fully meets the requirements, return only the function as code, with no additional
    explanations or comments.
```

Listing 18: System prompt for checking Python function validity (check_LLM).

```
Input/Output Format Description:
{format_description}
Function to be Evaluated:
{code}
```

Listing 19: User prompt for checking Python function validity (check_LLM).

## D.2 SYSTEM PROMPT: HINT_LLM

```
You are an expert in the domain of optimization heuristics. Your task is to offer practical hints to design
    better heuristics.
```

Listing 20: System prompt for hint generation (Hint_LLM).

### D.3 USER PROMPT: ARCHITECTURE HINTER

```
Please generate a hint focused on optimizing the {Optimization_Function} function, based on a deep
    understanding of its relationships and internal mechanisms with other functions.
{architecture_info}

[{Optimization_Function}]:
{Optimization_Function_code}
```

Listing 21: User prompt for architecture hinter function.

### D.4 SYSTEM PROMPT: GENERATOR_LLM

```
You are an expert in the domain of optimization heuristics. Your task is to design heuristics that can
    effectively solve optimization problems.
You are required to output Python code and nothing else. The output must strictly adhere to the following
    format:
```python
<your Python code>
```
```

Listing 22: System prompt for generating heuristic functions (Generator_LLM).

### D.5 USER PROMPT: COOPERATIVE HEURISTIC GENERATION

```
Explore and design a novel heuristic function '{Optimization_Function}' for {func_desc} based on the other
    functions and use the provided reflection to enhance the '{Optimization_Function}' function.
{Relationship_Description}

[{Function_code}]

[Reflection]
{reflection}

Output only the improved function in Python format, enclosed in a code block as follows:

```python
Your improved code
```
```

Listing 23: User prompt for cooperative heuristic generation (Generator_LLM).

