# OpenReview forum: "Unifying All Species: LLM-based Hyper-Heuristics for Multi-objective Optimization"
_ICLR.cc/2025/Conference — ICLR 2025 Conference Withdrawn Submission_

### Official Review · Reviewer_yB79 · 2024-10-25

**Soundness:** 1
**Presentation:** 1
**Contribution:** 2
**Rating:** 3
**Confidence:** 4

**Summary:**

This paper presents a novel framework called Multi-Objective Hierarchical Reflective Evolution (MHRE), which combines Large Language Models (LLMs) with hyper-heuristics to address multi-objective optimization problems. The MHRE framework significantly improves the ACO algorithm’s performance on random Traveling Salesman Problems (TSP) and TSPLib benchmark datasets. MHRE demonstrates superior performance in terms of optimization efficiency and adaptability when compared to traditional metaheuristic algorithms on a range of multi-objective optimization problems

**Strengths:**

1. The idea of leveraging Large Language Models (LLMs) to guide evolutionary algorithms is interesting and novel.
2. Figure 1 clearly illustrates the overall framework, highlighting the parts where LLMs are integrated into the process.
3. The appendix provides useful code and prompts, making it easier to understand the methodology.

**Weaknesses:**

1. The paper dedicates a significant amount of text to describing concepts, but the proposed method itself is not clearly explained. For example, Section 4 lacks sufficient details, with very few formulas or explanations to clarify how the method works.
2. The authors claim that GEMA integrates behaviors such as local and global search from evolutionary algorithms, but there is no detailed explanation of how this is achieved. Additionally, there are no concrete results presented to validate this claim.
3. The experiments are not comprehensive. For example, in Table 2, the traditional evolutionary algorithms selected are neither the most recent nor the strongest.

**Questions:**

1. In the GSES process, why is the LLM necessary for the standardization of the population? Would traditional methods be more effective for this step?
2. How does the proposed method ensure a balance between exploration and exploitation in the algorithm?
3. What exactly does the Architecture Upgrade do, and how does it ensure that the population evolves in the right direction?
4. To my knowledge, there are existing works that combine LLMs with evolutionary algorithms for multi-objective optimization. What are the key differences between the authors' approach and these existing works?

---

### Official Review · Reviewer_JcDj · 2024-10-31

**Soundness:** 1
**Presentation:** 1
**Contribution:** 1
**Rating:** 1
**Confidence:** 3

**Summary:**

This paper introduces the MHRE framework, which is designed to optimize and generate heuristics for multiobjective optimization problems. The performance of the MHRE framework is empirically evaluated using the TSPLib benchmark datasets and the random TSP problems.

**Strengths:**

1. The source code is available. (However, I found the contents of reevo.py in your provided supplementary code to be **identical to the code provided in the paper [1]**)
2. The motivation is novel.

[1] Ye, Haoran, et al. ‘Reevo: Large language models as hyper-heuristics with reflective evolution.’ arXiv preprint arXiv:2402.01145 (2024).

**Weaknesses:**

1. The clarity of the paper's writing may be improved to enhance understanding, as it currently presents akin to those encountered with **GPT-generated content**. (I think the definition of multi-objective has been changing throughout this article, and the proposed method is evaluated on the single-objective TSP benchmark while the motivation is solving multi-objective problems, so I find this article highly suspicious)
2. A more comprehensive review of relevant representation-related works, such as FunSearch[1] (which also belongs to LLM for Heuristic Design works), would enrich the discussion and provide additional context for the research.
3. Figures 1 and 2 lack explanations of the notations and details. Notions in Figure 2 are hard to comprehend and Figure 1 does not emphasize key points.
4. The problem definition could benefit from further clarification and refinement to ensure a clear and concise description for readers to fully grasp the research objectives and scope. Such as |H| in Definition 2.

[1] Romera-Paredes, Bernardino, et al. "Mathematical discoveries from program search with large language models." Nature 625.7995 (2024): 468-475.

**Questions:**

1. Why is the proposed method evaluated on the single-objective TSP benchmark while the motivation is solving multiobjective problems?
2. Since I found the contents of reevo.py in your provided supplementary code to be **identical to the code provided in the paper [1]**. Please give a detailed description on the difference between the proposed method and ReEvo.

---

### Official Review · Reviewer_Kuh7 · 2024-11-03

**Soundness:** 3
**Presentation:** 2
**Contribution:** 3
**Rating:** 5
**Confidence:** 3

**Summary:**

This paper proposes a novel framework, Multi-Objective Hierarchical Reflective Evolution (MHRE), for optimizing and generating heuristics algorithms for a broad range of optimization problems. The proposed methods show better performance on benchmark problems and TSP problems.

Overall, the motivations are clear and the ideas are good.

**Strengths:**

1. The motivations are clear and the proposed ideas are novel somehow.

2. The experiments are sufficient and the proposed method is better compared to several algorithms on different problems.

**Weaknesses:**

1. There are many grammatical errors and typos in the paper. I think it is not ready for publication. For example,

2. Figure 1 is not easy to understand.

3. The introduction of the comparison methods is needed.

4. The motivations for addressing the multiobjective optimization problem are strong, but the proposed approach is not specifically designed for multiobjective optimization.

5. The proposed method is not compared with state-of-the-art methods to show its superiority.

**Questions:**

In Tables 2-4, are there results from multiple runs? Given the generated methods are stochastic, it is suggested to report the mean results and the standard deviation.

1. Given the proposed method is able to solve multi-objective optimization problems, why the benchmark problems in Table 2 are single objective problems?

2. Please clarify the concept of heuristic and hyper-heuristic in the context of the paper.

3. How the proposed method addresses the multi-objective optimization problem/part is not well demonstrated. It seems that the proposed approach can be used for single and multi-objective optimization problems.

4. What are the sub-functions and architecture functions in the proposed method? In addition, what are the same type function, different-type function, and framework?

5. Why ACO is selected for TSP problems?

---

### Official Review · Reviewer_9BVQ · 2024-11-04

**Soundness:** 1
**Presentation:** 2
**Contribution:** 1
**Rating:** 1
**Confidence:** 5

**Summary:**

This paper introduces an approach, which uses LLMs to construct hyper-heuristics for optimization tasks. The heuristic is then assessed on traveling salesperson problems (TSP) and continuous single-objective optimization problems. For the TSP, the heuristic provides an alternative approach to ant-colony optimization (ACO) and is benchmarked on 26 small- to medium-sized instances of the TSPlib with instance sizes ranging between 48 and 1655 nodes. For continuous optimization, the heuristic is benchmarked against five algorithms from the infamous EC bestiary (https://github.com/fcampelo/EC-Bestiary).

**Strengths:**

The authors proposed a heuristic, which can be used to develop optimization algorithms for various optimization tasks. Moreover, the heuristic achieved performances that are at least comparable to the chosen competitors in their benchmark studies.

**Weaknesses:**

- The paper title emphasizes that the proposed heuristic works for multi-objective optimization, but throughout the paper, the authors only considered single-objective optimization tasks.
- For the benchmark studies, only rather simple optimization problems have been considered and the choice of competitors is not representative of the state of the art. In contrast, especially the algorithms that were considered in the continuous optimization problems are known to be part of the EC bestiary and should absolutely be avoided (see ref. [1, 2] listed below).
- For the TSP studies, the authors focussed on ACO. However, ACO is not a competitive TSP heuristic anymore. In fact, modern TSP approaches (e.g., refined versions of EAX and LKH) easily find the optimal tours for the TSP instances considered in Table 4.
- The first sentence of both the abstract and the introduction are identical, except for some missing words at the end of the sentence in the abstract.


[1] Camacho‐Villalón, Christian L., Marco Dorigo, and Thomas Stützle. "Exposing the grey wolf, moth‐flame, whale, firefly, bat, and antlion algorithms: six misleading optimization techniques inspired by bestial metaphors." International Transactions in Operational Research 30.6 (2023): 2945-2971.
DOI: 10.1111/itor.13176

[2] Aranha, Claus, Christian L. Camacho Villalón, Marco Dorigo, Felipe Campelo, Rubén Ruiz, Marc Sevaux, Kenneth Sörensen, and Thomas Stützle. "Metaphor-based metaheuristics, a call for action: the elephant in the room." Swarm Intelligence (2022) 16:1–6.
DOI: 10.1007/s11721-021-00202-9

**Questions:**

- Table 2 shows benchmark results for a small set of single-objective problems. However, I did not find a statement on how the performance is measured? Are the values showing the minimal values that the respective heuristics found per problem?
- Why did you consider particularly those ten benchmarking functions? There exist various tools and platforms, which allow to benchmark single-objective optimization heuristics across multiple benchmark functions, such as COCO, IOHprofiler, Nevergrad, etc.
- How many runs were performed for each algorithm per problem? And what was the termination criterion per algorithm (stagnation, fixed number of function evaluations, CPU time, ...)?
- There exist various heuristic approaches that are capable of achieving state-of-the-art performance across multiple optimization domains (including TSP and single-objective optimization). I suggest, the authors have a look at topics such as automated algorithm selection, automated algorithm configuration, and automated algorithm design.
- To assess the (reproducibility of the) results, I would encourage the authors to share their code.

---

### Note · Authors · 2024-12-03

I have read and agree with the venue's withdrawal policy on behalf of myself and my co-authors.